# The Contribution of New Breed Purple Wheat (8526-2 and 8529-1) Varieties Wholemeal Flour and Sourdough to Quality Parameters and Acrylamide Formation in Wheat Bread

Dovilė Klupsaite [1], Aura Kaminskaite [2], Arnoldas Rimsa [2], Agne Gerybaite [2], Agne Stankaityte [2], Ausra Sileikaite [2], Elzbieta Svetlauskaite [2], Emilija Cesonyte [2], Giedre Urbone [2], Karolis Pilipavicius [2], Konstancija Vaiginyte [2], Marija Vaisvilaite [2], Vilte Prokopenko [2], Giedre Stukonyte [2], Vytaute Starkute [1,2], Egle Zokaityte [1], Vita Lele [1,2], Darius Cernauskas [1,3], Ernestas Mockus [1], Zilvinas Liatukas [4], Vytautas Ruzgas [4], João Miguel Rocha [5,6] and Elena Bartkiene [1,2,*]

1  Institute of Animal Rearing Technologies, Lithuanian University of Health Sciences, Tilzes Str. 18, LT-47181 Kaunas, Lithuania
2  Department of Food Safety and Quality, Lithuanian University of Health Sciences, Tilzes g. 18, LT-47181 Kaunas, Lithuania
3  Food Institute, Kaunas University of Technology, Radvilenu Rd. 19, LT-50254 Kaunas, Lithuania
4  Institute of Agriculture, Lithuanian Research Centre for Agriculture and Forestry, Instituto al. 1, Akademija, LT-58344 Kedainiai, Lithuania
5  Laboratory for Process Engineering, Environment, Biotechnology and Energy, Faculty of Engineering, University of Porto, 4200-465 Porto, Portugal
6  Associate Laboratory in Chemical Engineering, Faculty of Engineering, University of Porto, 4200-465 Porto, Portugal
*  Correspondence: elena.bartkiene@lsmuni.lt; Tel.: +37-060-135-837

**Abstract:** The aim of this study was to evaluate the influence of purple wheat (varieties 8526-2 and 8529-1) wholemeal flour (PWWF) left untreated or fermented with *Lactiplantibacillus plantarum* No. 135 on the quality parameters of and acrylamide formation in wheat bread. Different quantities (5, 10, 15, and 20%) of PWWF were tested for bread preparation. Acidity, colour characteristics, hardness, enzyme activities, and antioxidant activity of PWWF, as well as bread quality and acrylamide concentration, were analysed. Differences in LAB count and amylolytic and proteolytic enzyme activities between two varieties of non-treated and fermented PWWF were not found. Fermentation increased DPPH-scavenging activity and reduced hardness of both PWWF varieties. A very strong positive correlation was found between total phenolic compound content and antioxidant activity in PWWF (r = 0.816, p = 0.001). In most cases, PWWF addition lowered bread specific volume and mass loss after baking. After 72 h of storage, bread with 5% PWWF showed the lowest hardness. Addition of 15% PWWF to bread increased overall acceptability. Fermentation and wheat variety significantly affected bread crumb a* colour coordinates. Addition of fermented PWWF significantly decreased acrylamide formation in bread ($p \leq 0.0001$), and bread with 5% PWWF variety 8526-1 had the lowest acrylamide content. In conclusion, the addition of new-breed PWWF to wheat bread improved certain quality parameters, while PWWF fermented with *L. plantarum* possessed DPPH-scavenging activity and can be recommended for acrylamide reduction in wheat bread.

**Keywords:** purple wheat; bread; fermentation; acrylamide; lactic acid bacteria; antioxidant activity

## 1. Introduction

Although wheat bread is one of the most popular and accepted foods around the world, its production from refined wheat flour shows a low functional value, as the main valuable phytochemicals are removed with the outer layer fraction [1,2]. In addition to enrichment of wheat bread with various functional ingredients (different pseudo cereals, pure phenolic compounds, dietary fibre of various origin, etc.), the application of coloured wheat for

higher-value bread production becomes very attractive. Purple wheat came into existence in the 19th century [3–5], and, nowadays, because its specific desirable phytochemical composition (high anthocyanin content) and health benefits, is receiving more and more attention [6]. Anthocyanins are located in the pericarp of the wheat grain [7], and the red colour of cereal grain is due to the presence of phlobaphenes in the outer layer [8]. Cyanidin-3-glucoside, cyanidin-3-(6-malonyl glucoside), cyanidin-3-rutinoside, peonidin-3-glucoside, and peonidin-3-(6-malonylglucoside) are the main anthocyanins in purple wheat grain [9]. Taking into consideration that the outer layer of wheat grain (especially in coloured wheat) is a valuable functional material for higher-value bread production, it would be beneficial to replace part of the refined wheat flour in bread with purple wheat grain wholemeal flour. However, the addition of cereal outer layer to the main bread formula is complicated.

Despite the many valuable compounds contained in cereal bran [10], dietary fibers show adverse effects on the properties of bread because of dilution of the gluten network, reduction of specific volume, etc. [11]. It has been suggested that the fermentation of cereal bran is necessary to attenuate the non-desirable effects of dietary fibre [12], and the application of sourdough starter culture can lead to production of high-quality bread enriched with desirable cereal bran compounds [13]. In this study, we hypothesized that coloured wheat wholemeal fermentation with *Lactiplantibacillus plantarum* No. 135 strain could be an appropriate technique to enrich wheat bread with coloured wheat grain wholemeal outer layer compounds, as well as to reduce acrylamide concentration in bread, because of the additional antioxidant properties of the coloured wheat wholemeal.

It has been reported that coloured wheat has much better antioxidant properties in comparison with traditional wheat grain [6]. The major antioxidants in coloured wheat are phenolic acids, flavones, flavanols, and anthocyanins [14]. In addition to the fact that the cereal outer layer exhibits desirable antioxidant characteristics [15], the antioxidants could reduce acrylamide formation in bread [16].

During bread production, the Maillard reaction leads to the formation of aroma, colour, and taste compounds; however, in the same reaction toxic substances, including acrylamide, are formed [16]. The main pathway for acrylamide formation in bread is the reaction of reducing sugars with asparagine [17–19]. Acrylamide is classified as a human carcinogen by the International Agency for Research on Cancer (IARC) [20,21]. The presence of acrylamide in daily consumed food (e.g., bread) have raised global concerns due to the fact that humans might be exposed to significant quantities of acrylamide in the long run [21–23]. Studies have been conducted on the effects of polyphenols, including phenolic extracts from plants or pure polyphenols, on acrylamide formation, and it was concluded that some polyphenols have significant potential to lower this toxic compound's formation [16]. Recently, many studies have reported on the supplementation of bread with various phenolics [24–27]. However, phenolic compounds can affect dough physicochemical and rheological properties, as well as bread-quality attributes, including volume, texture, and sensory characteristics [28–31]. Moreover, addition of pure extracts, which are not typical bread ingredients, can lead to a lower overall acceptability of the bread by consumers. In this study we hypothesized that coloured wheat wholemeal as a source of antioxidants, in combination with fermentation with a selected lactic acid bacteria (LAB) strain, can lead to acrylamide concentration reduction in bread.

The aim of this study was to evaluate the influence of non-treated and fermented with *L. plantarum* No. 135 purple wheat (varieties 8526-2 and 8529-1) wholemeal flour on the quality parameters and acrylamide formation in wheat bread. For this purpose, different quantities of non-fermented and fermented purple wheat wholemeal flour were tested for wheat bread preparation (5, 10, 15, and 20%). Non-fermented coloured wheat wholemeal flour and sourdough samples were analysed for acidity parameters (pH, total titratable acidity (TTA)), colour characteristics, hardness, amylolytic and proteolytic enzyme activities, total phenolic compound content, and free-radical-scavenging activity (DPPH)). Bread quality and safety characteristics (specific volume, crumb porosity, shape coefficient,

mass loss after baking, crust and crumb colour coordinates, overall acceptability, and acrylamide concentration) were also examined.

## 2. Materials and Methods

### 2.1. Materials Used for Bread Preparation

Wheat flour (type 550D, falling number 350 s, gluten 27%, ash 0.68%) obtained from 'Malsena plius' Ltd. mill (Panevezys, Lithuania) was used for the wheat bread (WB) preparation. The WB samples were prepared without and with addition of non-treated and fermented purple wheat wholemeal flour (5, 10, 15, and 20%). The *L. plantarum* No. 135 strain, showing versatile carbohydrate metabolism, tolerance to acidic conditions, and acrylamide reducing properties [32], was used for purple wheat wholemeal fermentation. Strain No. 135 was stored at −80 °C in a Microbank system (PRO-LAB DIAGNOSTICS) and propagated in an MRS broth (CM 0359; Oxoid Ltd., Hampshire, UK) at 30 °C for 48 h. Characteristics of the *L. plantarum* No. 135 strain are given in Table 1.

**Table 1.** Characteristics of the *Lactiplantibacillus plantarum* No. 135 strain.

| * 100 bp DNA-Ladder Extended | *Lactiplantibacillus plantarum* No. 135 | Gas Production, Tolerance to Temperature (10, 30, 37 and 45 °C) and Low pH Conditions (pH 2.5 for 2 h), and Carbohydrate Metabolism of the *Lactiplantibacillus plantarum* No. 135 Strain | | | |
|---|---|---|---|---|---|
|  |  | Glicerol | - | Esculin | +++ |
| | | D-arabinose | - | Salicin | +++ |
| | | L-arabinose | +++ | D-cellobiose | +++ |
| | | D-ribose | +++ | D-maltose | +++ |
| | | D-xylose | - | D-lactose | +++ |
| | | L-xylose | - | D-melibiose | +++ |
| | | D-adonitol | - | D-saccharose | +++ |
| | | Methyl-ßd-xYlopiranoside | - | D-trehalose | +++ |
| | | D-galactose | +++ | Inulin | - |
| | | D-glucose | +++ | D-melezitose | +++ |
| | | D-fructose | +++ | D-raffinose | - |
| | | D-mannose | +++ | Amidon | - |
| | | L-sorbose | - | Glycogen | - |
| | | L-rhamnose | + | Xylitol | - |
| | | Dulcitol | - | Gentiobiose | ++ |
| | | Inositol | - | D-turanose | +++ |
| | | D-mannitol | +++ | D-tagatose | +++ |
| | | D-sorbitol | +++ | D-fucose | - |
| | | Methyl-αD-mannopyranoside | +++ | L-fucose | - |
| | | Methyl-αD-glucopyranoside | + | D-arabitol | - |
| | | N-acetylglucosamine | +++ | L-arabitol | - |
| | | Amigdalin | +++ | Potassium gluconate | ++ |
| | | Arbutin | +++ | Potassium 2-ketogluconate | - |
| | | | | Potassium 5-ketogluconate | - |
| | | Gas production (+/-) | | - | |
| | | Temperature tolerance | | 10 °C | - |
| | | | | 30 °C | ++ |
| | | | | 37 °C | + |
| | | | | 45 °C | + |
| | | pH 2.5 | | 0 h $\log_{10}$ (CFU/mL) | $8.08 \pm 0.2$ |
| | | | | 2 h $\log_{10}$ (CFU/mL) | $7.69 \pm 0.1$ |

* Bands of the isolated LAB genus (analyzed by the BioNumerics v4.0 software package). Interpretation of lactic acid bacteria (LAB) growth in API 50 CH system and API 20 E system: +++ = strong growth (yellow); ++ = moderate growth (green); + = weak growth (dark green); − = no growth (blue); n.d. = not determined.

### 2.2. Characteristics of Purple Wheat Used in Experiment and Sourdough Preparation

Purple wheat (pedigree of purple wheat lines: DS8526-2 (PS Karkulka/RGT Reform), DS8529-1 (PS Karkulka/Delawaro)) was grown in the experiment conditions given below. The field experiment was designed in 4 replications (plot size 11 × 1.6 m), each replicate was grown in a separate block, with the field plots' arrangement randomised. The trials were conducted under intensive growing technology. The soil was light loam

*Endocalcari-Epihypogleyic-Cambisol*. Topsoil (0–30 cm) pH is slightly alkaline (7.5), medium in humus (2.4%), high in available phosphorus (253 mg/kg $P_2O_5$), and moderate in available potassium (142 mg/kg $K_2O$). Winter wheat was sown with treated seeds at a seed rate of 4.5 million/ha in the second part of September 2021 after the green-fallow. Complex mineral fertilisers ($N_{24}P_{66}K_{136}$) were applied to the whole experimental field before sowing. Nitrogen fertilisers (ammonium nitrate) were applied after resumption of spring vegetation (on 19 of April 2022) and when plants reached the stem elongation stage (on 16th May 2022). The rate of N100+60 was used. Weeds were controlled using the recommended herbicides in the autumn and spring. Yield was harvested on 4 August. The elevation of the experimental area is 82 m above sea level, and it belongs to the mid-latitude climate zone in the southwestern subregion of the Atlantic continental forest area. According to data from the local Dotnuva Meteorological Station (55°23′49.0″ N 23°51′55.0″ E, Figure 1), the climatic conditions were characterized by a long-term (1924–2022) average annual temperature of 6.5 °C and precipitation of 570 mm. The autumn of 2021 was slightly warmer (0.7 °C) and dryer than usual. The winters were quite mild, crop damage was very low. Spring–summer vegetation periods were cooler (1.3 °C) and very rainy (double precipitation rate). Very heavy rains partially washed away nitrogen and sulphur, thereby worsening the nutrition of plants. As well, plants roots were stressed by air deficiency in soil due to constantly wet soils. Wheat grain yield and quality were lower than usual in the rainy vegetation period.

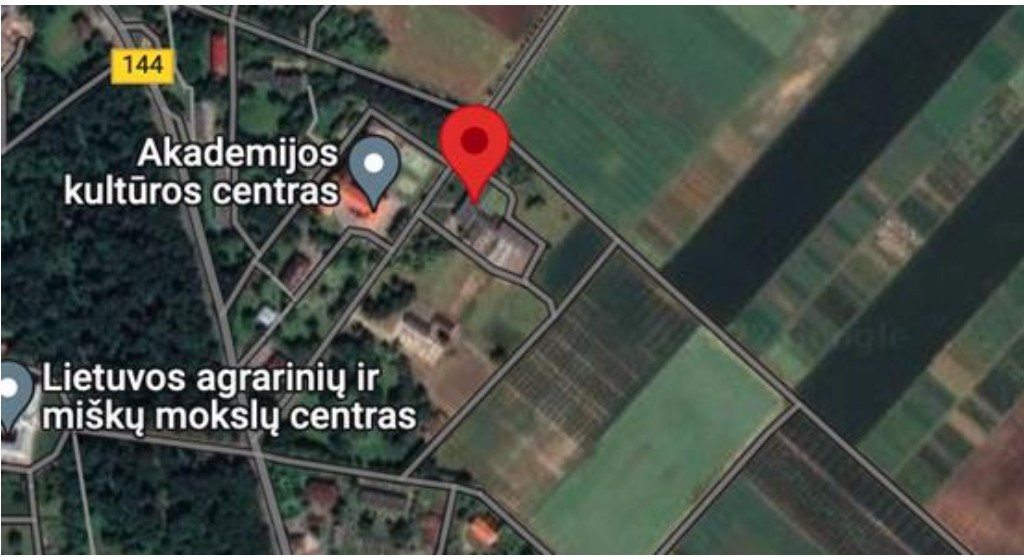

**Figure 1.** Location of the field experiment (Dotnuva, Lithuania).

For purple wheat fermentation, the *L. plantarum* No. 135 strain was used. The *L. plantarum* No. 135 cell suspension (5 mL), containing about 9.1 $\log_{10}$ CFU/mL, was added into the wholemeal and water mixture (50:50 by mass), followed by fermentation for 24 h at 30 °C. Non-fermented and sourdough samples prepared with 5, 10, 15, or 20% PWWF (of the total flour content) were used for bread production.

### 2.3. Bread Preparation

The WB formula consisted of 1 kg of wheat flour, 1.5% salt, 3% fresh compressed yeast, and 600 mL water (control bread). Control WB samples were prepared without the addition of purple wheat flour or sourdough. The tested sample groups were prepared by addition of 5, 10, 15, or 20% non-treated purple wheat flour and sourdough to the main recipe. In total 18 groups of dough and bread samples were prepared and tested. The dough was mixed for 3 min at a low speed, then for 7 min at high speed in a dough mixer (KitchenAid Artisan, Greenville, OH, USA). Then, the dough was left at 22 ± 2 °C for 10 min relaxation. After, the dough was shaped into 425 g loaves, formed, and proofed at 30 ± 2 °C and 80% relative humidity for 60 min. The bread was baked in a deck oven (EKA,

Borgoricco PD, Italy) at 220 °C for 25 min. The principle scheme of the experiment is shown in Figure 2.

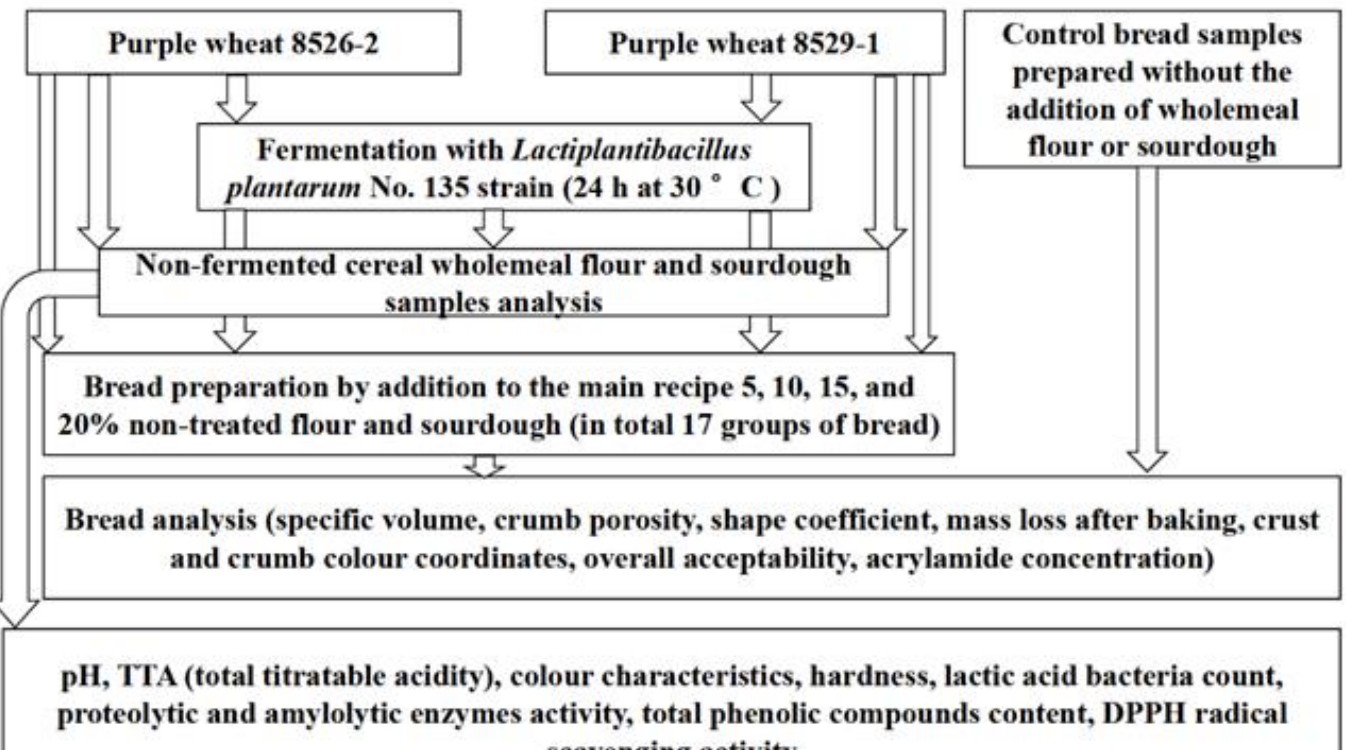

**Figure 2.** Principle scheme of the experiment.

*2.4. Evaluation of Non-Treated Purple Wheat Wholemeal Flour and Sourdough Parameters*

Non-treated and fermented purple wheat flour (after 0 and 24 h of fermentation) samples were analysed to evaluate their pH, TTA (total titratable acidity), LAB count, colour characteristics, hardness, free amino acid content, amylolytic and proteolytic enzyme activity, total phenolic compound content, and DPPH-radical-scavenging activity.

The pH value of samples was measured and recorded with a pH electrode (PP-15; Sartorius, Goettingen, Germany). For pH analysis, the electrode was immersed directly into a sourdough sample.

The total titratable acidity (TTA) was determined for a 10 g sample homogenized with 90 mL of distilled water and expressed as millilitres of 0.1 mol/L NaOH required to achieve a pH of 8.2.

For the evaluation of LAB count, 10 g of flour was homogenized with 90 mL of saline (9 g/L NaCl solution). Serial dilutions of $10^{-4}$ to $10^{-8}$ with saline were used for sample preparation. Sterile MRS (de Man, Rogosa, Sharpe) agar (CM0361, Oxoid, Fisher Scientific Ireland Ltd, Dublin, Ireland) of 5 mm thickness was used for bacterial growth on Petri plates. The plates were separately seeded with the sample suspension on the agar surface and were incubated under anaerobic conditions at 30 °C for 72 h. The number of bacterial colonies was calculated and expressed as a decimal $\log_{10}$ of colony-forming units per gram of sample (CFU/g).

Colour parameters were evaluated using a CIE L\*a\*b\* system (CromaMeter CR-400, Konica Minolta, Tokyo, Japan) [33].

Sample hardness was measured as the energy required for sample deformation (CT3 Texture Analyzer, Brookfield, WI, USA): 50 g of dough was placed in a Petri dish and compressed to 10% of its original height at a crosshead speed of 10 mm/s; the resulting peak energy of compression was reported as dough hardness.

The amylase activity was determined using the starch–iodine method as described by Nguyen et al. [34], with some modifications as described by Bartkiene et al. [35]. The dough sample (5 g) was homogenized with 50 mL of distilled water and centrifuged at $5000\times g$ for 10 min. The reaction mixture containing 1 mL of 1% (*w/v*) soluble starch as substrate in 1/15 M phosphate buffer at pH 6 and 0.5 mL sample extract was incubated for 10 min at 30 °C. The reaction was stopped, and the colour was developed by addition of 1.5 mL of diluted iodine reagent (a solution of iodine (2 mL, 0.25% *w/v*) with KI (2.5% *w/v*) diluted with 0.5 M HCl to 100 mL). Absorbance was measured at 670 nm using a Genesys 10 spectrophotometer (Thermo Fisher Scientific Inc., Langensenbold, Germany). One unit of α-amylase activity (1 AU) was defined as the amount of enzyme that catalyzes 1 g soluble starch hydrolysis to dextrins in 10 min at 30 °C.

The mode of action of the LAB protease was determined with a Sigma-Aldrich non-specific protease assay [36]. The dough sample (5 g) was dissolved in 50 mL of 10 mM sodium acetate buffer (pH 7.5) with 5 mM calcium acetate. For each sample 5 mL of 65% casein (*w/v*) as substrate and 1 mL of sample extract were incubated at 37 °C for 10 min. The reaction was stopped by the addition of 110 mM trichloroacetic acid (5 mL) and was maintained at 37 °C for 30 min. After centrifugation at $5000\times g$ for 10 min the supernatant (2 mL) was taken and added to 5 mL of 0.5 M sodium carbonate and 1 mL of Folin-Ciocalteu reagent (Sigma-Aldrich, Buchs, Switzerland). The protease activity was detected spectrophotometrically, since the released tyrosine developed a blue colouration. Each sample was read in a Genesys 10 spectrophotometer (Thermo Fisher Scientific Inc., Langensenbold, Germany) at 660 nm. One protease unit was defined as the amount of casein hydrolyzed to produce a colour equivalent to 1.0 μmol (181 μg) of tyrosine per minute at 37 °C and pH 7.5.

The total content of phenolic compounds (TPC) was determined using the spectrophotometric method, as reported by Vaher et al. [37]. A 0.2 mL of sample was combined with 0.8 mL of a saturated sodium carbonate solution and 1 mL of the Folin-Ciocalteu reagent. A spectrophotometer (J.P. SELECTA S.A. V-1100D, Barcelona, Spain) was used to measure the mixture's absorbance at 765 nm after it had stood at room temperature for 30 min. The total phenolic content of the sample was calculated by comparison with a standard curve produced by examining gallic acid.

The DPPH radical-scavenging assay was carried out using Zhu et al.'s [38] methodology. In short, the sample (2 mL), which was dissolved in the extraction solvent at various concentrations (0.4–1.2 mg/mL), and DPPH solution (2 mL, 0.1 mM, in ethanol) were combined. After shaking the reaction mixture for 60 min at room temperature in the dark, the absorbance was measured at 517 nm in comparison to a blank. The preparation of the controls was identical to that of the test group, with the exception that the appropriate extraction solvent was used in place of the antioxidant solution. The sample's ability to block the DPPH radical was estimated using the formula given in Zhu et al. [38].

### 2.5. Evaluation of Bread Quality Characteristics

After 12 h of cooling at $22 \pm 2$ °C, WB samples were subjected to analysis of specific volume, crumb porosity, shape coefficient, mass loss after baking, crust and crumb colour coordinates, sensory characteristics and overall acceptability, and acrylamide concentration.

Bread volume was established using the AACC method [39], and the specific volume was calculated as the ratio of volume to weight. Bread crumb porosity was evaluated with the LST method 1442:(1996) [40]. The bread-shape coefficient was calculated as the ratio of bread slice width to height (in mm). Mass loss after baking was calculated as a percentage by measuring loaf dough mass before baking and after baking. Crust and crumb colour parameters were evaluated using a CIE L*a*b* system (CromaMeter CR-400, Konica Minolta, Japan) [33]. Bread crumb hardness was determined as the energy required for sample deformation (CT3 Texture Analyzer, Brookfield, USA): bread slices of 2 cm thickness were compressed to 10% of their original height at a crosshead speed of 10 mm/s; the resulting peak energy of compression was reported as crumb hardness.

Three replicates from three different sets of baking were analysed and averaged. Sensory characteristics and overall acceptability of breads was carried out by 10 trained judges according to ISO method 8586-1 [41] using a 140 mm hedonic line scale ranging from 140 (extremely like) to 0 (extremely dislike).

### 2.6. Determination of Acrylamide in Bread

The acrylamide concentration was determined according to the method of Zhang et al. [42] with modification. The bread samples were homogenized in a blender (Ika A10, IKA®-Werke GmbH & Co. KG, Staufen, Germany). A 2 g sample was weighed in a 50 mL centrifuge tube and diluted with 20 mL of distilled/deionized water. The tube was vortexed (ZX3 Advanced VELP, VELP Scientifica Srl, Usmate Velate, Italy) for 10 min to mix the contents. The tube was centrifuged at 4000 rpm for 10 min with a centrifuge (Hermle Z 306, HERMLE Labortechnik GmbH, Wehingen, Germany). The ten millilitres of the clarified aqueous layer solution in 15 mL centrifuge tubes were clarified with 100 μL Carrez I (85 mM $K_4[Fe(CN)_6] \times 3H_2O$) and 100 μL Carrez II (250 mM $ZnSO_4 \times 7H_2O$) solutions. The tubes were centrifuged at 4000 rpm for 10 min. Acrylamide analytical standard solution (15.2 mg, 30.4 μg/L, 99.8% purity) was weighed and dissolved in a 1000 mL volumetric flask and diluted with deionized water. The obtained solution was diluted by pouring 2 mL of the acrylamide solution into a 1000 mL measuring flask and diluting it with deionized water. A 3 mL sample of the supernatant (or standard solution) was derivatized in a glass tube by adding 1.5 g of potassium bromide (KBr), 1 mL of potassium bromate solution (0.1 M, $KBrO_3$), and 0.3 mL of sulfuric acid solution (50%, $H_2SO_4$). The mixture was mixed in a shaker and kept for 2 h in a refrigerator (~4 °C). The derivative was neutralized by adding 250 μL of sodium thiosulphate solution (1 M, $Na_2S_2O_3 \times 5H_2O$) until the orange colour disappeared. About 1.5 g of sodium chloride (NaCl) was added to the derivatization mixture and the mixture was extracted with ethyl acetate ($CH_3COOC_2H_5$) (2 × 5 mL). The collected ethyl acetate was concentrated with a concentration system (Christ CT 02-50, Martin Christ, Osterode am Harz, Germany) at a temperature of 40 °C and reduced pressure. The solvent was evaporated and dissolved in 0.5 mL of ethyl acetate (for the standard, in a volume of 3 mL). The 100 mg of anhydrous sodium sulphate ($Na_2SO_4$) and 20 μL of triethylamine (($C_2H_5$)3N) (20 μL of triethylamine in 0.5 mL of a concentrated derivatization solution) was added to the solution in a 15 mL centrifuge tube, mixed, and centrifuged for 10 min (4000 rpm). The supernatant was analysed by GC–ECD. A gas chromatograph (Shimadzu GC-17A, Shimadzu Scientific Instruments, Kyoto, Japan) was equipped with an electron capture detector (ECD), an integrator to measure peak areas, and a thermostatted column. The capillary column such as Rxi-5Sil MS (Restek, Bellefonte, PA, USA) had the following properties: length 30 m; inner diameter 0.25 mm; stationary phase film thickness 0.25 μm. The working conditions were as follows: injection volume 1 μL; column temperature gradient 70 °C (hold 1 min), then 3 °C/min to 140 °C (hold 0.5 min), then 15 °C/min to 280 (hold 4 min); mobile phase nitrogen 18.0 cm/s flow rate, split 3.0; injector temperature 250 °C; detector temperature 260 °C; detector current 2 nA.

### 2.7. Statistical Analysis

The results were expressed as the mean values (for coloured wheat wholemeal and bread samples $n = 3$, and for bread overall acceptability $n = 10$ trained judges) ± standard error (SE). In order to evaluate the effects of different quantities of non-fermented and fermented coloured wheat grain wholemeal flour on dough and bread quality parameters, data were analysed with multivariate ANOVA (statistical program R 3.2.1, R Development Core Team., USA). Pearson correlations were calculated between coloured wheat wholemeal amount and bread quality and safety characteristics. The results were recognised as statistically significant at $p \leq 0.05$.

## 3. Results and Discussion

### 3.1. Parameters of the Non-Treated Purple Wheat Grain Wholemeal and Sourdough Samples

Acidity characteristics (pH and TTA), LAB count, colour coordinates, and hardness of non-treated and fermented purple wheat grain wholemeal samples are shown in Table 2, as well as enzyme activities in Table 3. Significant differences in pH were not found between non-treated and fermented wheat cereal varieties, and mean pH values were 5.59 in the non-treated group and 3.75 in the fermented samples. Although TTA did not differ significantly between fermented sample groups, the fermented 8529-1 variety showed 18.85% higher TTA than the fermented 8526-2 variety. Fermentation, wheat variety, and their interaction had significant effects on TTA (Table 4); however, significant correlations between pH and TTA were not found. These tendencies can be explained by the different wheat varieties' different buffering effects. It has been reported that the outer parts of the cereal grains contain compounds that act as buffering agents [43]. Higher quantities of the compounds with buffering characteristics in the fermentable substrate lead to organic acid neutralization, and in substrates with less buffering agents a low pH will be reached with lower organic acid production.

**Table 2.** Acidity characteristics (pH and total titratable acidity), lactic acid bacteria count, colour coordinates, and hardness of non-treated and fermented purple wheat wholemeal and sourdough samples.

| Samples | Acidity Parameters | | LAB Count, $\log_{10}$ CFU/g | Colour Characteristics | | | Texture Hardness, mJ |
|---|---|---|---|---|---|---|---|
| | pH | TTA, °N | | L* | a* | b* | |
| 8526-2 N-T | 5.48 ± 0.02 b | 3.22 ± 0.25 a | 4.80 ± 0.31 a | 44.27 ± 6.71 a | 6.71 ± 1.68 a | 12.13 ± 2.27 a | 0.30 ± 0.02 c |
| 8526-2 F | 3.80 ± 0.03 a | 9.17 ± 0.19 b | 8.44 ± 0.24 b | 58.05 ± 6.08 b | 9.96 ± 1.84 b | 16.50 ± 2.33 b | 0.10 ± 0.01 a |
| 8529-1 N-T | 5.70 ± 0.01 b | 3.51 ± 0.32 a | 4.10 ± 0.40 a | 51.39 ± 5.93 a | 5.41 ± 1.48 a | 10.83 ± 1.93 a | 0.20 ± 0.02 b |
| 8529-1 F | 3.70 ± 0.02 a | 11.33 ± 0.26 c | 8.32 ± 0.22 b | 61.47 ± 6.95 b | 8.30 ± 1.81 b | 14.62 ± 2.23 a | 0.10 ± 0.01 a |

8526-2 N-T—8526-2 non-treated purple wheat; 8526-2 F—8526-2 purple wheat fermented with *Lactiplantibacillus plantarum* No. 135 strain; 8529-1 N-T—8529-1 non-treated purple wheat; 8529-1 F—purple wheat fermented with *Lactiplantibacillus plantarum* No. 135 strain; N-T—non-treated; F—after 24 h of fermentation. TTA—total titratable acidity, LAB—lactic acid bacteria; CFU—colony-forming units; L* lightness; a* redness or −a* greenness; b* yellowness or −b* blueness. Data are expressed as mean values (*n* = 3) ± standard error (SE). a–c Mean values within a column with different letters are significantly different ($p \leq 0.05$).

**Table 3.** Proteolytic and amylolytic enzyme activities and antioxidant properties of the non-treated purple wheat grain wholemeal and sourdough samples.

| Samples | Enzyme Activities | | Antioxidant Properties | |
|---|---|---|---|---|
| | Proteolytic Enzyme Activity, PU | Amylolytic Enzyme Activity, AU | Total Phenolic Compound Content, mg/100 g d.m. | DPPH-Radical-Scavenging Activity, % |
| 8526-2 N-T | 139.8 ± 11.1 a | 152.4 ± 12.4 a | 52.9 ± 4.31 a | 6.10 ± 0.52 a |
| 8526-2 F | 145.9 ± 10.6 a | 160.9 ± 14.3 a | 110.8 ± 9.23 b | 56.9 ± 3.89 c |
| 8529-1 N-T | 141.7 ± 9.13 a | 145.1 ± 11.9 a | 120.4 ± 10.8 b | 33.4 ± 2.14 b |
| 8529-1 F | 150.5 ± 11.8 a | 166.9 ± 13.5 a | 101.1 ± 9.62 b | 65.4 ± 4.19 c |

8526-2 N-T—8526-2 non-treated purple wheat; 8526-2 F—8526-2 purple wheat fermented with *Lactiplantibacillus plantarum* No. 135 strain; 8529-1 N-T—8529-1 non-treated purple wheat; 8529-1 F—purple wheat fermented with *Lactiplantibacillus plantarum* No. 135 strain; N-T—non-treated; F—after 24 h of fermentation. Data are expressed as mean values (*n* = 3) ± standard error (SE). a–c Mean values within a column with different letters are significantly different ($p \leq 0.05$).

No significant different in LAB count were found between different varieties in the non-treated and fermented cereal wholemeal sample groups. Average LAB count in non-fermented samples was 4.45 $\log_{10}$ CFU/g, and in fermented ones, 8.38 $\log_{10}$ CFU/g. Usually, when mean LAB count in sourdough is $10^8$ CFU/g, sourdough is characterized as stable in terms of LAB domination and physical-chemical properties (e.g., acidification rate,

organic acid production, etc.) [44]. However, the changes over time should be controlled if this sourdough will be used further as a mother sourdough.

**Table 4.** Influence of analysed factors and their interaction on non-treated and fermented wheat wholemeal parameters.

| Bread Parameters | Factors and Their Interaction | | |
| --- | --- | --- | --- |
| | Non-Fermented/Fermented Wheat Wholemeal | Wheat Cereal Variety | Non-Fermented/Fermented Wheat Wholemeal × Wheat Cereal Variety |
| L* | **0.012** | 0.194 | 0.632 |
| a* | 0.979 | 0.647 | 0.868 |
| b* | 0.195 | 0.648 | 0.716 |
| Total phenolic compound content, mg/100 g d.m. | 0.074 | 0.530 | 0.717 |
| DPPH-radical-scavenging activity, % | 0.074 | 0.399 | 0.653 |
| Amylolytic enzyme activity, AU | 0.102 | 0.923 | 0.783 |
| Proteolytic enzyme activity, PU | 0.087 | 0.081 | 0.083 |
| LAB count, $\log_{10}$ CFU/g | 0.071 | 0.485 | 0.512 |
| pH | 0.394 | 0.976 | 0.938 |
| TTA, °N | **≤0.0001** | **≤0.0001** | **≤0.0001** |
| Texture hardness, mJ | 0.112 | 0.568 | 0.568 |

L*—lightness; a*—redness or −a*—greenness; b*—yellowness or −b*—blueness; TTA—total titratable acidity. The influence of analysed factors (fermentation and quantity of the scald) on bread parameters is significant when $p \leq 0.05$. Significant values are marked in bold.

For colour characteristics, higher L* (lightness) coordinates were found in fermented samples (both 8526-2 and 8526-1) than in non-treated samples (by 31.13 and 19.61%, respectively). Non-treated and fermented samples of 8526-2 had higher a* (redness) coordinates than 8526-1 by 19.37 and 16.67%, respectively. B* (yellowness) were higher in 8526-2 than in 8526-1 samples by 10.71 and 11.39% for non-treated and fermented samples, respectively. A moderate significant positive correlation was found between samples' L* and LAB count (r = 0.611, *p* = 0.035). Although there was no correlation between LAB count and TTA, there was a significant positive correlation between L* coordinate and TTA (r = 0.766, *p* = 0.004). These findings can be explained by the low stability of anthocyanins. By increasing TTA, anthocyanin content may have been reduced, leading to a higher L* coordinate of the fermentable substrate. Overall, the stability of anthocyanins is affected by temperature, pH, light, and composition of the fermentable substrate [45]. Moreover, this study showed that the wheat cereal varieties used in this experiment, had different buffering capacities, and that increasing LAB counts in substrate were related to a higher L* coordinate. However, no correlation between LAB count and TTA was found.

In both wheat varieties, fermentation reduced sample texture hardness, but no correlations were found between texture hardness and any of the analysed parameters (acidity, LAB count, amylolytic and proteolytic enzymes activity). The decrease in hardness of the wholemeal flour during fermentation can be explained by complex interactions, including competition for water by the water-soluble and water-insoluble fiber constituents, starch degradation, etc. The fiber, or non-starch polysaccharide, fraction, of whole wheat is composed primarily of arabinoxylans [46,47]. During fermentation, these compounds are broken down into lower-molecular-weight substances, and these changes can lead to a lower-hardness fermentable substrate.

Significant differences in proteolytic and amylolytic enzyme activities were not found between the different wheat varieties and treatments (Table 3). Proteolytic enzyme activity was, on average, 144.3 PU, and average amylolytic enzyme activity was 156.3 AU. Wheat flour is a raw material, which, in addition to the main constituents, also includes a variety of enzymes. However, microorganisms are considered the main source of enzymes because their reproduction rate is high and they excrete bioactive compounds, including various enzymes, into the fermentable substrate [48]. In wheat flour, α-amylase activity is low,

and β-amylases are abundant but have little or low activity [48]. The influence of cereal fermentation is associated with organic acid synthesis, activation of the flour endogenous enzymes, and microbial secondary metabolic activity [49–51]. In addition, LAB possess a variety of enzymatic activities [52–54], although this characteristic is strain-specific [55,56]. This study found no significant differences between the amylolytic and proteolytic enzyme activities of fermented and non-treated purple wheat wholemeal samples.

The lowest TPC content occurred in non-fermented 8526-2 samples (52.9 mg/100 g d.m.); however, after fermentation TCP was increased, on average, by 52.3% (Table 3). Significant differences in TPC content between non-treated and fermented 8526-2 samples were not found, and TPC content was, on average, 110.8 mg/100 g d.m.

In all cases fermentation increased antioxidant DPPH-radical-scavenging activity of the samples—in 8526-2 samples, on average, by 9.32 times, and in 8526-1 samples, on average, by 1.95 times (Table 3). A very strong positive correlation between TPC content and DPPH-radical-scavenging activity was found (r = 0.816, *p* = 0.001).

It has been reported that cereals and cereal-based products contain significant levels of antioxidants [57,58], and fermentation of cereals can enhance these properties as well [59]. However, although fermentation has a positive influence on TPC and antioxidative activity of cereals, the degree of influence depends on the microorganism used to treat the cereal grain [60]. Đorđević et al. [59] reported no correlation between TPC content and DPPH-radical-scavenging activity in cereals. However, this study was performed with traditional wheat (*Triticum durum*), and, according to Brand-Williams, Cuvelier, and Berset [61], ferulic acid, the main phenolic acid in traditional wheat grains, showed a weak antiradical effect in experiments with the DPPH radical, which may explain the discrepancies. However, in our study the tested coloured wheat antioxidant properties can be associated with other compounds (pigments, e.g., anthocyanins), and this can explain the different results obtained and the correlation between the coloured wheat grain wholemeal TPC content and DPPH. Finally, the use of fermentation as a separate process can enhance the levels of antioxidants in coloured wheat grain wholemeal and can be used to improve its functional properties.

### 3.2. Bread Quality Characteristics

Bread specific volume, porosity, shape coefficient, mass loss after baking, and bread crumb images are shown in Table 5. The highest bread specific volume occurred in the control bread samples, 8526-2 N-T-5% and 8526-2 F-15% (on average, 2.92 cm$^3$/g). Bread with 5% of the non-treated 8526-2 showed a 6.25% higher specific volume than 8526-2 F-5% samples. However, significant differences between bread samples prepared with 10% of the non-treated and fermented 8526-2 were not found. Fermentation led to a 24.0% higher specific volume of the bread enriched with 15% of 8526-2 sourdough compared to bread prepared with non-treated 8526-2 wholemeal. However, opposite tendencies were found in the bread enriched with 20% of 8526-2 sourdough: both non-treated and fermented 8526-2 addition at 20% led to significant bread specific volume reduction. Bread prepared with both non-treated and fermented 8529-1 wheat wholemeal had higher specific volume at 5, 10, and 15% addition levels, but lower at 20%.However, the analysed factors and their interactions did not significantly affect bread specific volume (Table 6).

Samples 8526-2 N-T-15%, 8526-2 N-T-20%, 8529-1 N-T-5%, 8529-1 N-T-10%, 8529-1 N-T-15%, 8529-1 N-T-20%, 8529-1 F-5%, and 8529-1 F-10% showed, on average, 8.76% lower porosity than samples 8526-2 N-T-5%, 8526-2 N-T-10%, 8526-2 F-5%, 8526-2 F-10%, 8526-2 F-15%, 8526-2 F-20%, 8529-1 F-15%, and 8529-1 F-20%. No analysed factors had significant effects on bread porosity (Table 6).

By increasing the non-treated 8526-2 wheat wholemeal quantity in the main bread formula, the shape coefficient of the bread was reduced; however, the addition of 15% 8526-2 sourdough increased the bread shape coefficient compared to samples prepared with 5 and 10%. Samples with non-treated 8529-1 wholemeal showed higher shape coefficients than samples prepared with the same quantities of 8529-1 wholemeal sourdough.

However, the analysed factors and their interactions did not significantly affect the bread shape coefficient (Table 6).

**Table 5.** Bread specific volume, porosity, shape coefficient, mass loss after baking, and bread crumb images.

| Bread Samples | Specific Volume, cm³/g | Porosity, % | Shape Coefficient | Mass Loss after Baking, % |
|---|---|---|---|---|
| Control | 2.91 ± 0.09 d | 74.51 ± 3.70 b | 2.31 ± 0.12 c,d | 16.50 ± 0.91 h |
| 8526-2 N-T-5% | 2.88 ± 0.12 d | 70.46 ± 3.50 b | 2.35 ± 0.15 c,d | 12.09 ± 0.79 e,f |
| 8526-2 N-T-10% | 2.50 ± 0.13 c | 68.43 ± 2.73 a,b | 2.42 ± 0.13 d | 10.97 ± 0.51 e |
| 8526-2 N-T-15% | 2.25 ± 0.08 b,c | 65.74 ± 2.30 a | 2.10 ± 0.15 c | 10.84 ± 0.60 d,e |
| 8526-2 N-T-20% | 2.10 ± 0.13 b | 63.20 ± 3.79 a | 1.66 ± 0.11 a,b | 14.60 ± 0.28 h |
| 8526-2 F-5% | 2.70 ± 0.15 c,d | 68.86 ± 2.40 a,b | 1.73 ± 0.10 b | 13.93 ± 0.68 f |
| 8526-2 F-10% | 2.50 ± 0.13 c | 67.53 ± 3.30 a,b | 1.75 ± 0.09 b | 6.70 ± 0.48 b |
| 8526-2 F-15% | 2.96 ± 0.13 d | 70.16 ± 4.20 a,b | 2.74 ± 0.16 e | 11.85 ± 0.58 e |
| 8526-2 F-20% | 1.66 ± 0.12 a | 68.86 ± 2.40 a,b | 1.63 ± 0.14 a,b | 13.93 ± 0.63 g |
| 8529-1 N-T-5% | 2.20 ± 0.07 b | 63.62 ± 3.68 a | 2.09 ± 0.12 c | 8.30 ± 0.52 c |
| 8529-1 N-T-10% | 2.30 ± 0.08 b,c | 61.25 ± 2.70 a | 2.22 ± 0.12 c | 9.06 ± 0.57 d |
| 8529-1 N-T-15% | 2.20 ± 0.07 b | 60.23 ± 3.31 a | 2.25 ± 0.14 c | 5.35 ± 0.32 a |
| 8529-1 N-T-20% | 2.36 ± 0.14 b,c | 60.11 ± 3.91 a | 2.07 ± 0.11 c | 13.35 ± 1.05 f |
| 8529-1 F-5% | 2.43 ± 0.13 b,c | 64.53 ± 3.23 a | 1.94 ± 0.10 b,c | 13.84 ± 1.11 f |
| 8529-1 F-10% | 2.49 ± 0.09 c | 63.47 ± 2.22 a | 1.45 ± 0.09 a | 11.86 ± 0.59 e |
| 8529-1 F-15% | 2.34 ± 0.12 b,c | 68.31 ± 3.97 a,b | 1.65 ± 0.12 a,b | 16.02 ± 0.93 i |
| 8529-1 F-20% | 2.22 ± 0.12 b | 67.80 ± 2.98 a,b | 1.73 ± 0.12 b | 10.23 ± 0.73 d |

| Control | 8526-2 N-T-5% | 8526-2 N-T-10% | 8526-2 N-T-15% | 8526-2 N-T-20% |

| 8526-2 F-5% | 8526-2 F-10% | 8526-2 F-15% | 8526-2 F-20% | 8529-1 N-T-5% |

| 8529-1 N-T-10% | 8529-1 N-T-15% | 8529-1 N-T-20% | 8529-1 F-5% | 8529-1 F-10% |

| 8529-1 F-15% | | | 8529-1 F-20% |

8526-2 N-T—8526-2 non-treated purple wheat; 8526-2 F—8526-2 purple wheat fermented with *Lactiplantibacillus plantarum* No. 135 strain; 8529-1 N-T—8529-1 non-treated purple wheat; 8529-1 F—purple wheat fermented with *Lactiplantibacillus plantarum* No. 135 strain; Control—bread prepared without purple wheat flour or sourdough; NT—bread with non-treated purple wheat flour; F—bread with purple wheat flour sourdough. 5%, 10%, 15%, 20%—bread prepared with 5%, 10%, 15%, 20%, respectively, non-treated purple wheat flour or sourdough. Data are expressed as mean values (*n* = 3) ± standard error (SE). a–i Mean values within a column with different letters are significantly different ($p \leq 0.05$).

**Table 6.** Influence of analysed factors and their interactions on bread specific volume, porosity, shape coefficient, and mass loss after baking.

| Bread Parameters | Factors and Their Interaction | | | | | | |
|---|---|---|---|---|---|---|---|
| | Non-Fermented/ Fermented Wheat Wholemeal | Wheat Cereal Variety | Non-Fermented/ Fermented Wheat Wholemeal Quantity | Non-Fermented/ Fermented Wheat Wholemeal × Wheat Variety | Non-Fermented/ Fermented Wheat Wholemeal × Non-Fermented/ Fermented Wheat Wholemeal Quantity | Wheat Variety × Non-Fermented/ Fermented Wheat Wholemeal Quantity | Wheat Variety × Non-Fermented/ Fermented Wheat Wholemeal Quantity × Non-Fermented/ Fermented Wheat Wholemeal |
| Specific volume, cm$^3$ g$^{-1}$ | 0.446 | 0.482 | 0.485 | 0.928 | 0.858 | 0.882 | 0.996 |
| Porosity, % | 0.187 | 0.141 | 0.343 | 0.542 | 0.857 | 0.360 | 0.819 |
| Shape coefficient | 0.621 | 0.386 | 0.952 | 0.254 | 0.575 | 0.134 | 0.439 |
| Mass loss after baking, % | **0.032** | 0.538 | 0.414 | **0.028** | 0.088 | 0.395 | 0.636 |

Influence of analysed factors (fermentation and quantity of the scald) on bread parameters is significant when $p \leq 0.05$. Significant values are marked in bold.

The highest mass lost after baking was showed by control samples and samples prepared with 15% 8529 wholemeal sourdough. Fermentation, as well as the interaction of non-fermented/fermented wheat wholemeal and wheat variety, significantly affected bread mass after baking ($p = 0.032$ and $p = 0.028$, respectively) (Table 6).

Bread production is a very complex process and the measurement of rheological parameters assists in controlling the behaviour of dough and the quality of the final bread [62]. The inclusion of various types of dietary fibers, e.g., wholemeal flour, in bread production significantly influences both processing and quality of bread [62]. Dietary fibers can interfere with protein association, weaken the dough, and affect gelling and pasting [63]. There is a wide range of studies on the physical parameters of bread with dietary fibers. In most cases, negative effects of dietary fibers on bread volume and moisture loss were observed and attributed to gluten dilution and lower gas retention [62,64–66]. A similar tendency was also found in our study. Moreover, the degree of substrate breakdown, acidification properties, and metabolism of LAB affect the quality of bread made with sourdough. The drop in pH due to LAB activity during sourdough fermentation causes swelling of gluten and arabinoxylans as well as hydrolysis of starch [67]. This also accelerates the activity of not only LAB proteolytic enzymes, which induce gluten proteolysis, but also endogenous cereal enzymes [50]. This is significant for bread volume, gas retention, and dough rheology. Reduction of gluten viscosity during prolonged fermentation as well as increased degree of softening and lower resistance to extension were observed in fermented doughs [67]. However, other studies reported improved bread texture of wheat bread with sourdough [67]. Other compounds produced by LAB during fermentation, such as exopolysaccharides, glucose, mannitol, and acetate, could be related to increased loaf volume, water absorption of the dough, and delayed bread staling [10]. According to Sun et al. *L. plantarum* is usually chosen due to its ability to significantly elicit both decline and expansion in the hardness, cohesiveness, and viscoelasticity of whole-wheat bread [68]. In our study, the pH was lower in bread with both wheat wholemeal varieties fermented with *L. plantarum*, but proteolytic and amylolytic enzyme activities were similar in both non-treated and fermented samples. This could partly contribute to the observed changes in bread quality attributes.

*3.3. The Changes of Bread Texture Hardness during the Storage*

After 12 h of storage, samples prepared with 5% non-treated 8526-2 wholemeal showed lower hardnesscompared to control samples (Figure 3).

Similar tendencies were found in the samples prepared with 5% non-treated 8526-1 wholemeal: their hardness was the same as that of controls (0.5 mJ). However, by increasing non-treated and fermented coloured wheat wholemeal content in the main bread formula, bread hardness was increased.

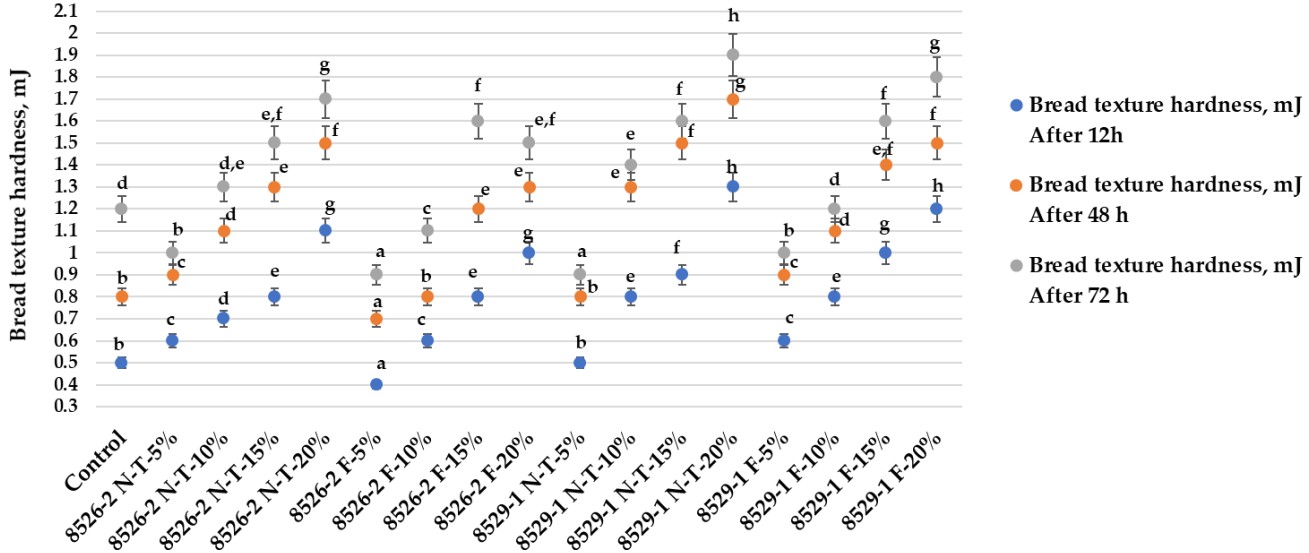

**Figure 3.** Bread texture hardness (mJ) after 24, 48, and 72 h of storage (8526-2 N-T—8526-2 non-treated purple wheat; 8526-2 F—8526-2 purple wheat fermented with *L. plantarum* No. 135 strain; 8529-1 N-T—8529-1 non-treated purple wheat; 8529-1 F—purple wheat fermented with *L. plantarum* No. 135 strain; Control—bread prepared without purple wheat flour or sourdough; NT—bread with non-treated purple wheat flour; F—bread with purple wheat flour sourdough. 5%, 10%, 15%, 20%—bread prepared with 5%, 10%, 15%, and 20%, respectively, non-treated purple wheat flour or sourdough. Data are expressed as mean values ($n$ = 3) ± standard error (SE). a–h Mean values within all samples after the same storage period with different letters are significantly different ($p \leq 0.05$)).

After 48 h of storage, most of the samples showed higher hardness in comparison with control breads, except for 8526-2 F-5%, whose hardness was, on average, 12.5% lower, and 8526-2 F-10% and 8529-1 N-T-5%, whose hardness did not differ from the control value of 0.8 mJ.

After 72 h of storage, samples 8526-2 N-T-5%, 8526-2 F-5%, 8526-2 F-10%, 8529-1 N-T-5%, and 8529-1 F-5% showed lower hardness than that of control samples. However, the analysed factors and their interactions did not significantly affect bread hardness during storage (Table 7).

**Table 7.** Influence of analysed factors and their interaction on bread texture hardness after 24, 48, and 72 h of storage.

| Bread Parameters | | Factors and Their Interaction | | | | | | |
|---|---|---|---|---|---|---|---|---|
| | | Non-Fermented/Fermented Wheat Wholemeal | Wheat Cereal Variety | Non-Fermented/Fermented Wheat Wholemeal Quantity | Non-Fermented/Fermented Wheat Wholemeal × Wheat Variety | Non-Fermented/Fermented Wheat Wholemeal × Non-Fermented/Fermented Wheat Wholemeal Quantity | Wheat Variety × Non-Fermented/Fermented Wheat Wholemeal Quantity | Wheat Variety × Non-Fermented/Fermented Wheat Wholemeal Quantity × Non-Fermented/Fermented Wheat Wholemeal |
| Texture hardness, mJ | After 24 h | 0.795 | 0.460 | 0.137 | 0.736 | 0.994 | 0.990 | 0.994 |
| | After 48 h | 0.217 | 0.954 | 0.508 | 0.657 | 0.655 | 0.747 | 0.664 |
| | After 72 h | 0.715 | 0.338 | 0.471 | 0.567 | 0.806 | 0.593 | 0.790 |

Influence of analysed factors (fermentation and quantity of the scald) on bread parameters is significant when $p \leq 0.05$.

Bread staling is caused by the retrogradation of amylose and amylopectin [69]. Dietary fibers in wholemeal could merge these compounds in order to slow down the staling process, but the effect mainly depends on the type, content, and particle size of dietary

fibers [70,71]. The addition of DF induces denser and firmer texture in bread [72]. A higher content of insoluble fiber in bread could increase bread firmness during storage [73] and this tendency was also observed in our study. A number of studies reported that dietary fibers improved the shelf-life of bread, while the application of sourdough when baking with wheat has been found to have mixed effects on the shelf-life of wheat bread [74]. The positive effect of purple wheat sourdough on the delay of wheat bread staling could be explained by the production of certain metabolites and the enzymatic activity of LAB. Exopolysaccharides act as hydrocolloids and result in greater water retention and softer crumb structure, while organic acids enhance amylase and protease activities, thus decreasing the staling rate [75].

### 3.4. Bread Overall Acceptability, Crust and Crumb Colour Coordinates and Acrylamide Concentration

Bread overall acceptability is shown in Figure 4, crust and crumb colour coordinates are given in Table 8 and acrylamide concentration is shown in Figure 5. In most cases, addition of non-treated and fermented coloured wheat cereal grain wholemeal at 15% to the main bread formula increased overall bread acceptability, in comparison with the control group and groups prepared with 5, 10, and 20% of non-treated and fermented coloured wheat cereal grain wholemeal (Figure 4). Significant differences in overall acceptability were not found between non-fermented and fermented groups with the same quantity of wheat wholemeal added, and the analysed factors and their interactions did not significantly affect overall bread acceptability (Table 9).

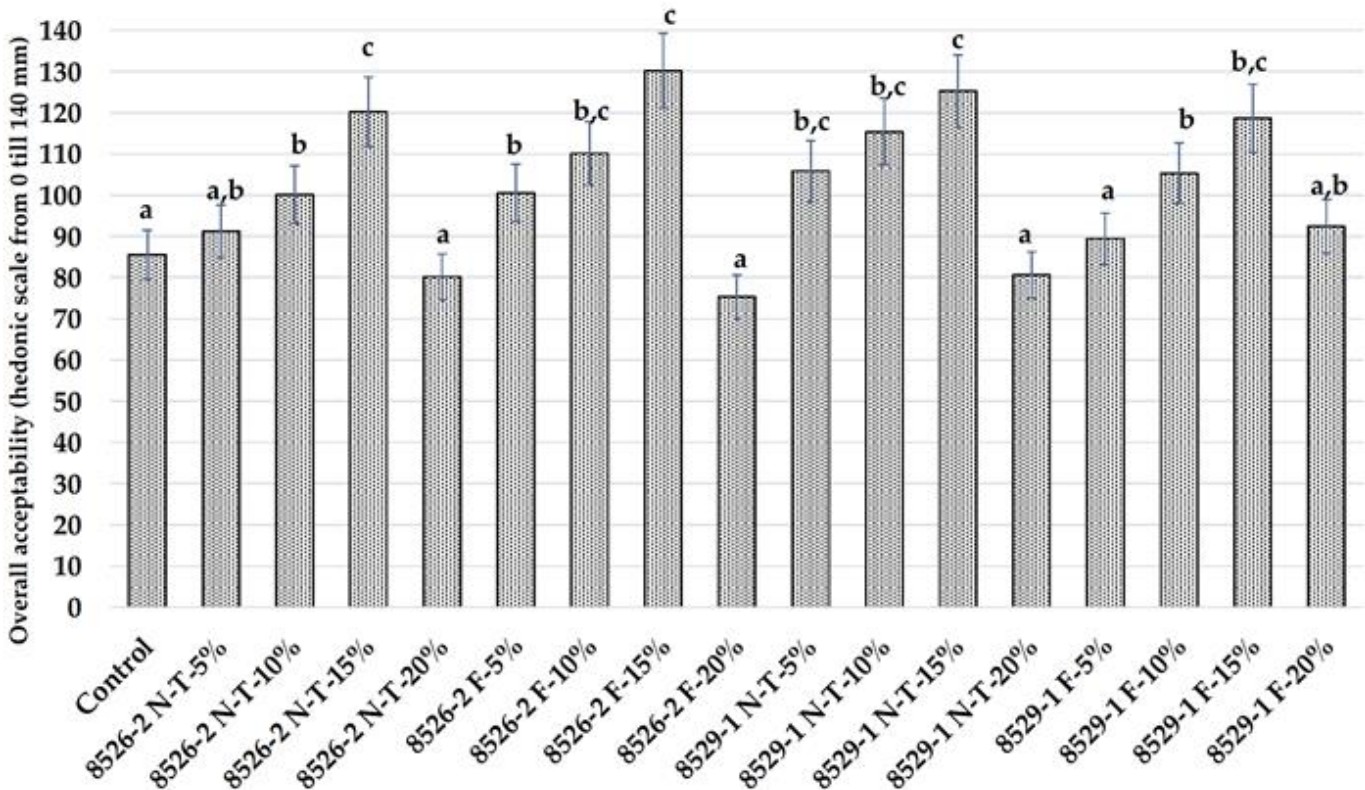

**Figure 4.** Bread overall acceptability (Control—bread prepared without purple wheat flour or sourdough; NT—bread with non-treated purple wheat flour; F—bread with purple wheat flour sourdough. 5%, 10%, 15%—bread prepared with 5%, 10%, 15%, respectively, non-treated purple wheat flour or sourdough. Data are expressed as mean values ($n = 10$) $\pm$ standard error (SE). a–c Mean values within all samples with different letters are significantly different ($p \leq 0.05$)).

**Table 8.** Colour characteristics of the bread crust and crumb.

| Bread Samples | Crust | | | Crumb | | |
|---|---|---|---|---|---|---|
| | L* | a* | b* | L* | a* | b* |
| Control | 49.09 ± 2.69 a | 10.74 ± 0.26 a | 19.81 ± 0.77 a | 77.71 ± 4.58 a | 0.17 ± 0.04 a | 22.21 ± 1.31 a |
| 8526-2 N-T-5% | 54.15 ± 3.85 b | 12.10 ± 0.46 b | 24.62 ± 1.06 b | 77.85 ± 3.59 a | 0.27 ± 0.01 b | 21.32 ± 1.22 a |
| 8526-2 N-T-10% | 58.33 ± 3.21 b | 12.32 ± 0.51 b | 23.14 ± 1.13 b | 77.45 ± 2.71 a | 0.28 ± 0.04 b | 22.12 ± 1.13 a |
| 8526-2 N-T-15% | 58.40 ± 2.34 b | 12.41 ± 0.46 b | 23.83 ± 0.96 b | 77.52 ± 3.03 a | 0.31 ± 0.03 b | 22.22 ± 1.19 a |
| 8526-2 N-T-20% | 66.74 ± 2.67 c | 12.24 ± 0.36 b | 23.75 ± 1.31 b | 76.24 ± 3.43 a | 0.34 ± 0.01 b,c | 21.21 ± 1.20 a |
| 8526-2 F-5% | 57.02 ± 2.28 b | 12.35 ± 0.69 b | 24.42 ± 1.47 b | 76.39 ± 4.77 a | 0.95 ± 0.04 f | 21.29 ± 1.29 a |
| 8526-2 F-10% | 60.88 ± 3.64 b | 12.81 ± 0.43 b | 24.59 ± 1.2 b | 77.71 ± 3.03 a | 0.96 ± 0.03 f | 21.63 ± 1.53 a |
| 8526-2 F-15% | 63.44 ± 2.67 b,c | 13.42 ± 0.69 b | 23.19 ± 1.16 b | 76.02 ± 3.42 a | 0.98 ± 0.02 f | 21.66 ± 1.36 a |
| 8526-2 F-20% | 65.00 ± 2.31 c | 13.53 ± 0.44 b,c | 23.58 ± 1.04 b | 76.31 ± 3.74 a | 0.99 ± 0.03 f | 22.57 ± 1.01 a |
| 8529-1 N-T-5% | 59.00 ± 3.42 b | 11.59 ± 0.49 b | 24.18 ± 1.43 b | 77.14 ± 3.28 a | 0.48 ± 0.03 d | 21.73 ± 1.15 a |
| 8529-1 N-T-10% | 60.46 ± 2.41 b | 11.93 ± 0.65 b | 24.26 ± 0.95 b | 76.09 ± 3.75 a | 0.47 ± 0.03 d | 22.70 ± 1.15 a |
| 8529-1 N-T-15% | 63.93 ± 2.11 b,c | 12.02 ± 0.32 b | 24.17 ± 1.33 b | 76.10 ± 3.42 a | 0.49 ± 0.04 d | 22.11 ± 1.12 a |
| 8529-1 N-T-20% | 64.48 ± 3.01 b,c | 12.51 ± 0.46 b | 25.15 ± 1.21 b | 76.01 ± 2.62 a | 0.52 ± 0.05 d,e | 22.26 ± 1.13 a |
| 8529-1 F-5% | 58.43 ± 3.49 b | 12.29 ± 0.59 b | 22.23 ± 1.17 b | 78.63 ± 3.95 a | 0.28 ± 0.02 b | 21.05 ± 1.16 a |
| 8529-1 F-10% | 60.51 ± 2.39 b | 12.30 ± 0.63 b | 23.58 ± 1.19 b | 79.65 ± 4.87 a | 0.44 ± 0.03 d | 22.32 ± 1.22 a |
| 8529-1 F-15% | 64.97 ± 1.93 b,c | 13.90 ± 0.48 c | 23.10 ± 0.98 b | 79.93 ± 4.56 a | 0.58 ± 0.04 d,e | 22.91 ± 1.17 a |
| 8529-1 F-20% | 65.21 ± 2.11 c | 13.51 ± 0.68 b,c | 24.25 ± 1.16 b | 79.70 ± 4.14 a | 0.62 ± 0.03 e | 22.89 ± 1.34 a |

8526-2 N-T—8526-2 non-treated purple wheat; 8526-2 F—8526-2 purple wheat fermented with *L. plantarum* No. 135 strain; 8529-1 N-T—8529-1 non-treated purple wheat; 8529-1 F—purple wheat fermented with *L. plantarum* No. 135 strain; Control—bread prepared without purple wheat flour or sourdough; NT—bread with non-treated purple wheat flour; F—bread with purple wheat flour sourdough. 5%, 10%, 15%—bread prepared with 5%, 10%, 15%, respectively, non-treated purple wheat flour or sourdough. L* lightness; a* redness or −a* greenness; b* yellowness or −b* blueness; Data are expressed as mean values (*n* = 3) ± standard error (SE). a–f Mean values within a column with different letters are significantly different (*p* ≤ 0.05).

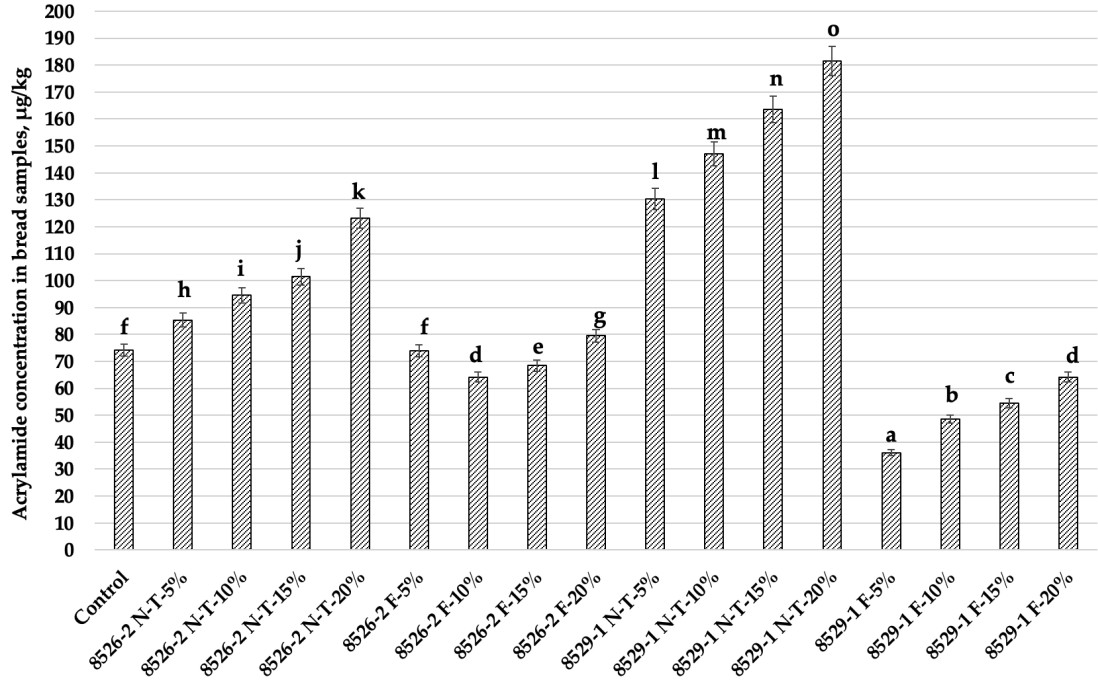

**Figure 5.** Acrylamide concentration (µg/kg) in bread samples (8526-2 N-T—8526-2 non-treated purple wheat; 8526-2 F—8526-2 purple wheat fermented with *L. plantarum* No. 135 strain; 8529-1 N-T—8529-1 non-treated purple wheat; 8529-1 F—purple wheat fermented with *L. plantarum* No. 135 strain; Control—bread prepared without purple wheat flour or sourdough; NT—bread with non-treated purple wheat flour; F—bread with purple wheat flour sourdough. 5%, 10%, 15%—bread prepared with 5%, 10%, 15%, respectively, non-treated purple wheat flour or sourdough. Data are expressed as mean values (*n* = 3) ± standard error (SE). a–o Mean values with different letters are significantly different (*p* ≤ 0.05).

**Table 9.** Influence of analysed factors and their interaction on acrylamide concentration in bread.

| Bread Parameters | | Non-Fermented/ Fermented Wheat Wholemeal | Wheat Cereal Variety | Non-Fermented/ Fermented Wheat Wholemeal Quantity | Non-Fermented/ Fermented Wheat Wholemeal × Wheat Variety | Non-Fermented/ Fermented Wheat Wholemeal × Non-Fermented/ Fermented Wheat Wholemeal Quantity | Wheat Variety × Non-Fermented/ Fermented Wheat Wholemeal Quantity | Wheat Variety × Non-Fermented/ Fermented Wheat Wholemeal Quantity × Non-Fermented/ Fermented Wheat Wholemeal |
|---|---|---|---|---|---|---|---|---|
| Overall acceptability | | 0.539 | 0.459 | 0.865 | 0.390 | 0.760 | 0.856 | 0.712 |
| | L* | 0.769 | 0.622 | 0.281 | 0.284 | 0.215 | 0.524 | 0.601 |
| Crust | a* | 0.342 | 0.252 | 0.548 | 0.289 | 0.639 | 0.434 | 0.927 |
| | b* | 0.377 | 0.071 | 0.165 | 0.339 | 0.632 | 0.166 | 0.693 |
| | L* | 0.750 | 0.672 | 0.321 | 0.379 | 0.810 | 0.827 | 0.113 |
| Crumb | a* | **≤0.0001** | **0.003** | 0.119 | **≤0.0001** | 0.247 | **0.048** | 0.054 |
| | b* | 0.521 | 0.388 | 0.402 | 0.462 | 0.895 | 0.500 | 0.871 |
| Acrylamide concentration | | **≤0.0001** | 0.412 | 0.252 | **0.006** | 0.139 | **0.035** | **0.037** |

L* lightness; a* redness or −a* greenness; b* yellowness or −b* blueness; Influence of analysed factors (fermentation and quantity of the scald) on bread parameters is significant when $p \leq 0.05$. Significant factors are marked in bold.

Similar to our results, other studies also reported that additions of such dietary fibers sources as wheat bran to bread formula had a positive effect on sensory properties of baked products [62,76–78]. Sourdough bread contains a greater amount of volatile compounds and that could result in higher scores in sensory tests [79]. According to the results of our study, by adding a higher quantity of non-treated or fermented purple wheat sourdough to bread, its acceptability was enhanced. The use of sourdough in bread making enhances texture and flavor attributes, leading to better consumer acceptance [80]. Acidification, protein hydrolysis, and release of phenolic compounds during sourdough fermentation contribute to bread flavor formation [68]. Furthermore, fermentation with LAB can reduce bitterness in breads prepared with wholemeal flour and increase fruitiness taste [81]. Mantzourani et al. reported that sourdough fermented with *Lacticaseibacillus paracasei* K5 enhanced bread sensory properties and acceptability to consumers [82].

Colour characteristics of the bread crust and crumb are shown in Table 8. In all the cases, by including coloured wheat wholemeal to the main bread formula, bread crust L*, a*, and b* coordinates were increased. Significant differences between samples in the crumb L* coordinates were not found; however, in all the cases, bread prepared with the addition of the coloured wheat wholemeal (non-treated and fermented) showed higher crumb a* coordinates than control breads, and, by increasing coloured wheat wholemeal quantities in the main bread formula, bread crumb a* coordinates were increased. However, significant differences in bread crumb b* coordinates were not established. It was found that fermentation and wheat variety, as well as wheat variety × non-fermented/fermented wheat wholemeal quantity interaction significantly affected bread crumb a* coordinates ($p \leq 0.0001$, $p = 0.003$, and $p = 0.0048$, respectively) (Table 9).

Colour is one of the essential attributes of the bread's quality because consumers are more likely to accept bread with a golden-brown crust and a creamy white crumb. It has been reported that dietary fibre compounds reduce bread crust lightness by increasing brown colour, due to an oxidation reaction, more intensive caramelization, and a higher amount of the accumulated melanoids during baking [83,84]. However, we obtained opposite results, which could be explained by the lighter colour of non-treated and fermented purple wheat grain wholemeal and sourdough samples. Moreover, the colour of the crumb more directly reflects the ingredients used for bread making, and a smaller size of dietary fibers could cause lower differences in colour compared to that of bread without it [85]. Similarly, in our study, the lightness of bread crumb did not differ between all bread samples. Higher values of the a* coordinate in breads prepared with the addition of coloured wheat wholemeal were due to the presence of anthocyanins and phlobaphenes in the cereal grain [8]. Furthermore, simple sugars generated at the end of sourdough fermentation and

LAB-induced release of phenolic compounds and anthocyanins could contribute to the change in colour coordinates of tested breads [86].

Acrylamide concentration in bread samples is given in Figure 5. Bread with coloured wheat wholemeal added showed higher concentrations of acrylamide, and, by increasing wheat wholemeal in the main bread formula, acrylamide concentration was increased. The highest acrylamide content occurred in samples prepared with 20% non-treated wheat wholemeal (8526-2 N-T-20%—123.1 µg/kg and 8529-1 N-T-20%—181.5 µg/kg). However, bread samples prepared with coloured wheat wholemeal sourdough showed significantly lower acrylamide content than breads prepared with non-treated coloured wheat wholemeal. Bread samples prepared with 8526-2 wholemeal at 5, 10, 15, and 20%, had 13.3, 32.1, 32.5, and 35.4 µg/kg lower acrylamide content, respectively. Bread samples prepared with 8526-1 wholemeal at 5, 10, 15, and 20%, had 72.3, 67.0, 66.7, and 64.6 µg/kg lower acrylamide content, respectively. Fermentation was a significant factor for acrylamide formation in bread ($p \leq 0.0001$). Significant effects on acrylamide concentration were found for the following interactions: non-fermented/fermented wheat wholemeal × wheat variety; wheat variety × non-fermented/fermented wheat wholemeal quantity; wheat variety × non-fermented/fermented wheat wholemeal quantity × non-fermented/fermented wheat wholemeal (Table 9). However, significant correlations between the bread crust and crumb colour coordinates and acrylamide concentration were not found.

Acrylamide is an unfavorable Maillard-reaction-derived compound with a potential neurotoxic and carcinogenic effect [87]. According to the European Commission, the set value for wheat-based bread is 80 µg/kg [88]. High contents of asparagine and reducing sugars, as well as baking temperaturse in the range of 140–180 °C, are optimal conditions for the formation of acrylamide [89]. Antioxidants could either inhibit or enhance acrylamide formation, but results in the literature are inconsistent [87]. Some authors reported that the oxidized forms of antioxidants (flavonoids and phenolic acids) inhibit acrylamide formation [77,78,87,90,91]. In this study, the acrylamide concentration of all breads made with non-treated coloured wholemeal wheat that showed antioxidant activity was higher than control breads. This could be explained by the fact that the free amino acid (e.g., asparagine) concentration in wholemeal flour is higher due to the presence of the outer layers of grain [92]. However, the lowest content of acrylamide was found in breads made with purple wheat wholemeal sourdough, which also possessed higher DPPH-scavenging activity, compared to non-treated purple wheat. During fermentation, LAB (especially *L. plantarum*) can excrete antioxidant-active compounds including active peptides and phenolics [10]. It has been reported that a lower content of acrylamide in baked goods could be related to glucose metabolism by LAB and reduced concentration of asparagine during sourdough fermentation [77]. This effect occurs due to the metabolism of such microorganisms as yeast and lactic acid bacteria, which utilize this amino acid for their growth [93]. Moreover, the reduction in pH during fermentation is also important because it lowers the reactivity of free asparagine (increases protonation of the amino acid) and further inhibits the formation of Schiff base, a precursor of acrylamide [94,95].

## 4. Conclusions

Fermentation with *L. plantarum* No. 135 increased the lightness and reduced the hardness of purple wheat (varieties 8526-2 and 8529-1) wholemeal flour. Proteolytic and amylolytic enzyme activities in the two varieties were similar, as were LAB counts and pH. Fermentation increased the DPPH-scavenging activity of purple wheat flour and it strongly correlated with the total phenolic compound content (r = 0.816, $p$ = 0.001). Breads with purple wheat flour had a reduced specific volume and mass loss after baking. During storage, bread hardness increased with the increased quantity of purple wheat flour in the bread formula. Wheat variety and fermentation had significant effects on bread crumb a* coordinates. Bread with 5% purple wheat flour showed the lowest hardness after 72 h of storage, while the addition of 15% of these flours (both varieties, non-treated and fermented) led to increased overall acceptability. By increasing the purple wheat flour content in the

main bread formula, the acrylamide concentration was increased and fermentation was a significant factor for acrylamide formation in bread ($p \leq 0.0001$). The lowest acrylamide concentration was found in bread with 5% purple wheat flour (8526-1). Incorporation of fermented purple wheat wholemeal flour could have potential advantages for wheat bread quality and acrylamide reduction.

**Author Contributions:** Conceptualization, E.B.; methodology, E.M., E.B., V.R. and J.M.R.; validation, D.K., V.S., E.M. and E.Z.; formal analysis, A.K., A.R., A.G., A.S. (Agne Stankaityte), A.S. (Ausra Sileikaite), E.S., E.C., G.U., K.P., K.V., M.V., V.P., G.S., V.S., E.Z., D.C., E.M., Z.L., V.P. and V.L.; investigation, A.K., A.R., A.G., A.S. (Agne Stankaityte), A.S. (Ausra Sileikaite), E.S., E.C., G.U., K.P., K.V., M.V., V.P., G.S., V.S., E.Z., D.C., E.M., Z.L., V.P. and V.L.; resources, E.B., V.R. and Z.L.; data curation, E.B.; writing—original draft preparation, E.B., D.K., V.S., E.Z. and D.C.; writing—review and editing, E.B., V.R., D.K. and J.M.R.; visualization, E.Z. and D.C.; supervision, E.B.; project administration, E.B. All authors have read and agreed to the published version of the manuscript.

**Funding:** This research received no external funding.

**Institutional Review Board Statement:** Not applicable.

**Informed Consent Statement:** Not applicable.

**Data Availability Statement:** Not applicable.

**Acknowledgments:** The authors gratefully acknowledge the COST Action 18101 SOURDOMICS—Sourdough bio-technology network towards novel, healthier and sustainable food and bioprocesses (https://sourdomics.com/; https://www.cost.eu/actions/CA18101/, accessed on 17 November 2022), where the author E.B. is the Vice-Chair and leader of the working group 6 "Project design and development innovative prototypes of products and small-scale processing technologies", and the author J.M.R. is the Chair and Grant Holder Scientific Representative, and is supported by COST (European Cooperation in Science and Technology) (https://www.cost.eu/, accessed on 17 November 2022). COST is a funding agency for research and innovation networks. Regarding the author J.M.R., this work was also financially supported by LA/P/0045/2020 (ALiCE) and UIDB/00511/2020–UIDP/00511/2020 (LEPABE) funded by national funds through FCT/MCTES (PIDDAC).

**Conflicts of Interest:** The authors declare no conflict of interests.

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
