# Peer review of "The Contribution of New Breed Purple Wheat (8526-2 and 8529-1) Varieties Wholemeal Flour and Sourdough to Quality Parameters and Acrylamide Formation in Wheat Bread"

_fermentation, doi:10.3390/fermentation8120724_

Round 1

Reviewer 1 Report

The manuscripts entitled “The contribution of new breed purple wheat (8526-2 and 8529-1) varieties wholemeal flour and sourdough to quality parameters and acrylamide formation in wheat bread” reported that the acidity, colour characteristics, hardness, enzymes activities, and antioxidant activity of fermented purple (PWWF), as well as bread quality and acrylamide concentration. It is interesting topic and match the aims and scopes of fermentation.

1.       Why authors choose the Varieties 8526-2 and 8529-1 as material. What advantages between 8526-2, 8529-1 and others varieties.

2.       Line 172, authors described samples were analysed to evaluate their pH, TTA (total titratable acidity), LAB count, colour characteristics, hardness, free amino acids content, amylolytic and proteolytic enzymes activity, total phenolic compounds content, and DPPH radical scavenging activity. Please provided experimental details of DPPH radical scavenging ability. Whether the antioxidant activity of Line 218-219 means DPPH scavenge? I think authors exaggerated this experimental result, because DPPH is only an antioxidant index in vitro. Additionally, authors should carefully check the accuracy of References, Line 219 (References 38) should an evaluated method of antioxidant activity, rather than the saponin induced apoptosis by caspase-8-dependent cleavage of Bcl-2.

3.       Line 217, authors should provide experimental details of measuring total content of phenolic compounds (TPC). Additionally, I suggested that authors should measure the contents of anthocyanin before and after fermentation.

4.       It is interesting that fermentation decreased the acrylamide concentration. Authors should highlight this result in manuscript and pay more discussion about this result.

Author Response

Reviewer 1: The manuscripts entitled “The contribution of new breed purple wheat (8526-2 and 8529-1) varieties wholemeal flour and sourdough to quality parameters and acrylamide formation in wheat bread” reported that the acidity, colour characteristics, hardness, enzymes activities, and antioxidant activity of fermented purple (PWWF), as well as bread quality and acrylamide concentration. It is interesting topic and match the aims and scopes of fermentation.

Reviewer 1: Why authors choose the Varieties 8526-2 and 8529-1 as material. What advantages between 8526-2, 8529-1 and others varieties.

Authors response: These varieties were chosen because they were the new breed lines of the purple wheat that were created at the Institute of Agriculture, Lithuanian Research Centre for Agriculture and Forestry. These varieties differ from other wheats (conventional i.e. red and white wheat) in a significantly higher amount of anthocyanins. Compared to blue wheat, purple wheat possesses a more diverse set of anthocyanins and their total amount is higher. In purple wheat, anthocyanins are concentrated on the surface of the grain (pericarp), so it is much easier to separate this layer than in the case of blue wheat, where anthocyanins are concentrated deeper (aleurone). Moreover, used purple wheat varieties are more productive than blue and black wheat.

Reviewer 1: Line 172, authors described samples were analysed to evaluate their pH, TTA (total titratable acidity), LAB count, colour characteristics, hardness, free amino acids content, amylolytic and proteolytic enzymes activity, total phenolic compounds content, and DPPH radical scavenging activity. Please provided experimental details of DPPH radical scavenging ability.

Authors response: Experimental details of DPPH radical scavenging ability was provided:

The DPPH radical scavenging assay was carried out using Zhu et al (2010) methodology. In a short, the sample (2 ml), which was dissolved in the extraction solvent at various concentrations (0.4-1.2 mg/ml), and DPPH solution (2 ml, 0.1 mM, in ethanol) were combined. After shaking the reaction mixture for 60 minutes at room temperature in the dark, the absorbance was measured at 517 nm in comparison to a blank. The preparation of the controls was very identical to that of the test group, with the exception that the appropriate extraction solvent was used in place of the antioxidant solution. The sample's ability to block the DPPH radical was estimated using the formula given in Zhu et al (2010).

Reviewer 1: Whether the antioxidant activity of Line 218-219 means DPPH scavenge? I think authors exaggerated this experimental result, because DPPH is only an antioxidant index in vitro. Additionally, authors should carefully check the accuracy of References, Line 219 (References 38) should an evaluated method of antioxidant activity, rather than the saponin induced apoptosis by caspase-8-dependent cleavage of Bcl-2.

Authors response: Yes, antioxidant activity of Line 218-219 means DPPH scavenge. The DPPH test is one of the most important tests used to determine the antioxidant activity and it is commonly applied to assess the antioxidant activity of plant extracts. 

“Antioxidant activity” in text was changed to the “DPPH radical scavenging activity”.

The reference for methodology of DPPH radical scavenging activity was revised: Zhu KX, Lian CX, Guo XN, Peng W, Zhou HM. Antioxidant activities and total phenolic contents of various extracts from defatted wheat germ. Food Chem. 2011;126:1122–1126. doi: 10.1016/j.foodchem.2010.11.144.

Reviewer 1: Line 217, authors should provide experimental details of measuring total content of phenolic compounds (TPC). Additionally, I suggested that authors should measure the contents of anthocyanin before and after fermentation.

Authors response: Authors are thankful for the valuable suggestion; however, during the conversion of dietary fibers, the release of phenolic compounds could be observed. Therefore, in this experiment, the total content of phenolic compounds (TPC) and DPPH radical scavenging activity were analysed.

Experimental details of measuring total content of phenolic compounds (TPC) were provided:

“0.2 ml of sample was combined with 0.8 ml of a saturated sodium carbonate solution and 1 ml of the Folin-Ciocalteau reagent. Spectrophotometer (J.P. SELECTA S.A. V-1100D, Barcelona, Spain) was used to measure the mixture's absorbance at 765 nm after it had stood at room temperature for 30 minutes. The total phenolic content of sample was calculated by comparison with a standard curve produced by examining gallic acid.”

Reviewer 1: It is interesting that fermentation decreased the acrylamide concentration. Authors should highlight this result in manuscript and pay more discussion about this result.

Authors response: the discussion was improved:However, the lowest content of acrylamide was found in breads made with purple wheat wholemeal sourdough, which also possessed higher DPPH scavenging activity, compared to non-treated purple wheat. During fermentation, LAB (especially L. plantarum) could excrete antioxidant-active compounds including active peptides and phenolics [10]. It was reported that the lower content of acrylamide in baked goods could be related with the glucose metabolism by LAB and reduced concentration of asparagine during sourdough fermentation [77]. This effect occurs due to the metabolism of such microorganisms as yeast and lactic acid bacteria, which utilize this amino acid for their growth [93]. Moreover, the reduction in pH during fermentation is also important because it lowers the reactivity of free asparagine (increases protonation of amino acid) and further inhibits the formation of Schiff base, a precursor of acrylamide [94,95].”

Reviewer 2 Report

This manuscript is very interesting and important, both from the point of view of fermented food production (use of new raw material and study of quality characteristics) and from the nutritional point of view (influence of the use of lactic acid bacteria to reduce undesirable components in products). Therefore, in my opinion, this manuscript should be accepted for publication in the FERMENTATION journal, albeit with previous corrections and additions. In order to improve the manuscript, my suggestions are as follows:

·         Section „2.3. Bread preparation” – please add information on the number of times the entire experiment was repeated.

·         Line 181 – I suggest you correct „Man, Rogosa, Sharpe” to „de Man, Rogosa, Sharpe”.

·         Line 209 – I suggest you correct „Folin-Ciocalteus” to „Folin-Ciocalteu”.

·         Lines 216-219 – please describe the methods at least in general terms, what they consisted of, what their principle was.

·         Section „Results and Discussion” – In my opinion, the discussion of the results obtained must focus more on the microorganisms tested and their physiological and biochemical properties. The Authors should look for correlations between the phenomena observed in the obtained bread samples and the microorganisms activity. This is extremely important when considering the thematic scope of the journal to which this manuscript has been submitted. What properties of Lactiplantibacillus plantarum No. 135 may have influenced the recorded quality parameters and acrylamide formation in wheat bread? How did these properties determine in both bread samples (i.e. with 8526-2 F and 8526-2 purple wheat).

·         Figure 3. – in my opinion, the dots presenting the results cannot be connected by lines, as they are independent samples.

Author Response

Reviewer 2: This manuscript is very interesting and important, both from the point of view of fermented food production (use of new raw material and study of quality characteristics) and from the nutritional point of view (influence of the use of lactic acid bacteria to reduce undesirable components in products). Therefore, in my opinion, this manuscript should be accepted for publication in the FERMENTATION journal, albeit with previous corrections and additions. In order to improve the manuscript, my suggestions are as follows:

Reviewer 2: Section „2.3. Bread preparation” – please add information on the number of times the entire experiment was repeated.

Authors response: From one baking batch of each tested group (17 groups: control bread; breads with 5, 10, 15 and 20% of non-treated purple wheat flour or sourdough of two varieties), 5 loaves were obtained, which were further used for non-destructive and destructive analysis, so in general 3 loaves (n=3) were used for each analysis.

Reviewer 2: Line 181 – I suggest you correct „Man, Rogosa, Sharpe” to „de Man, Rogosa, Sharpe”.

Authors response: corrected: „Man, Rogosa, Sharpe” was changed to „de Man, Rogosa, Sharpe”.

Reviewer 2: Line 209 – I suggest you correct „Folin-Ciocalteus” to „Folin-Ciocalteu”.

Authors response: corrected: „Folin-Ciocalteus” was changed to „Folin-Ciocalteu”.

Reviewer 2: Lines 216-219 – please describe the methods at least in general terms, what they consisted of, what their principle was.

Authors response: methods were described:

The total content of phenolic compounds (TPC) was determined by spectrophotometric method, as reported by Vaher et al. [37]. 0.2 ml of sample was combined with 0.8 ml of a saturated sodium carbonate solution and 1 ml of the Folin-Ciocalteu reagent. Spectrophotometer (J.P. SELECTA S.A. V-1100D, Barcelona, Spain) was used to measure the mixture's absorbance at 765 nm after it had stood at room temperature for 30 minutes. The total phenolic content of sample was calculated by comparison with a standard curve produced by examining gallic acid.

The DPPH radical scavenging assay was carried out using Zhu et al. [38] methodology. In a short, the sample (2 ml), which was dissolved in the extraction solvent at various concentrations (0.4-1.2 mg/ml), and DPPH solution (2 ml, 0.1 mM, in ethanol) were combined. After shaking the reaction mixture for 60 minutes at room temperature in the dark, the absorbance was measured at 517 nm in comparison to a blank. The preparation of the controls was very identical to that of the test group, with the exception that the appropriate extraction solvent was used in place of the antioxidant solution. The sample's ability to block the DPPH radical was estimated using the formula given in Zhu et al. [38].

Reviewer 2:    Section „Results and Discussion” – In my opinion, the discussion of the results obtained must focus more on the microorganisms tested and their physiological and biochemical properties. The Authors should look for correlations between the phenomena observed in the obtained bread samples and the microorganisms activity. This is extremely important when considering the thematic scope of the journal to which this manuscript has been submitted. What properties of Lactiplantibacillus plantarum No. 135 may have influenced the recorded quality parameters and acrylamide formation in wheat bread? How did these properties determine in both bread samples (i.e. with 8526-2 F and 8526-2 purple wheat).

Authors response: The discussion was improved:

3.2. Bread quality characteristics

“…..Moreover, the degree of substrate breakdown and acidification properties and metabolism of LAB affect the quality of bread made with sourdough. The drop in pH due to LAB activity during sourdough fermentation causes swelling of gluten and arabinoxylans as well as hydrolysis of starch [67]. This also accelerates the activity of not only LAB proteolytic enzymes, which induce gluten proteolysis, but also endogenous cereal enzymes [50]. This is significant for bread volume, gas retention, and dough rheology. Reduction of gluten viscosity during prolonged fermentation as well as increased degree of softening and lower resistance to extension were observed in fermented doughs [67]. However, other studies reported improved bread texture of wheat bread with sourdough [67]. Other compounds produced by LAB during fermentation, such as exopolysaccharides, glucose, mannitol, and acetate, could be related to increased loaf volume, water absorption of the dough, and delayed bread staling [10]. According to Sun et al. L. plantarum is usually chosen due to ability to significantly elicit both decline and expansion in the hardness, cohesiveness, and viscoelasticity of whole-wheat bread [68]. In our study, the pH was lower in fermented with L. plantarum wheat wholemeal of both varieties, but proteolytic and amylolytic enzyme activities were similar in both nontreated and fermented samples and this could partly contribute to the observed changes in bread quality attributes.”

3.3. The changes of bread texture hardness during the storage

“….The positive effect of purple wheat sourdough on the delay of wheat bread staling could be explained by the production of certain metabolites and the enzymatic activity of LAB. Exopolysaccharides act as hydrocolloids and result in greater water retention and softer crumb structure, while organic acids enhance amylase and protease activities, thus decreasing the staling rate [75].”

3.4. Bread overall acceptability, crust and crumb colour coordinates and acrylamide concentration

“…..Sourdough bread contains a greater amount of volatile compounds and that could result in higher scores in sensory tests [79]. According to the results of our study, by adding a higher quantity of nontreated or fermented purple wheat sourdough to bread, its acceptability was enhanced. The use of sourdough in bread making enhances texture and flavor attributes, leading to better consumer acceptance [80]. Acidification, protein hydrolysis, and release of phenolic compounds during sourdough fermentation contribute to the bread flavor formation [68]. Furthermore, fermentation with LAB can reduce bitterness in breads prepared with wholemeal flour and increase fruitiness taste [81]. Mantzourani et al. reported that sourdough fermented with Lacticaseibacillus paracasei K5 enhanced bread sensory properties and acceptability by consumers [82].”

“……Furthermore, simple sugars generated at the end of sourdough fermentation and LAB induced release of phenolic compounds and anthocyanins could contribute to the change in colour coordinates of tested breads [86].”

“…….However, the lowest content of acrylamide was found in breads made with purple wheat wholemeal sourdough, which also possessed higher DPPH scavenging activity, compared to non-treated purple wheat. During fermentation, LAB (especially L. plantarum) could excrete antioxidant-active compounds including active peptides and phenolics [10]. It was reported that the lower content of acrylamide in baked goods could be related with the glucose metabolism by LAB and reduced concentration of asparagine during sourdough fermentation [77]. This effect occurs due to the metabolism of such microorganisms as yeast and lactic acid bacteria, which utilize this amino acid for their growth [93]. Moreover, the reduction in pH during fermentation is also important because it lowers the reactivity of free asparagine (increases protonation of amino acid) and further inhibits the formation of Schiff base, a precursor of acrylamide [94,95].”

Reviewer 2:  Figure 3. – in my opinion, the dots presenting the results cannot be connected by lines, as they are independent samples.

Authors response: Lines were deleted and figure 3 was revised:

Round 2

Reviewer 1 Report

I think this article is acceptable.

Reviewer 2 Report

I see that the Authors have conscientiously revised the manuscript taking into account the reviewer's comments.

In my opinion, the current version of the manuscript is suitable for publication, is in line with the scope of the Fermentation journal and presents valuable results of the Authors' own research and the constructive discussion.

I have no further comments or objections to this manuscript, I accept it in its current version for publication.